# A Small Molecule That Promotes Cellular Senescence Prevents Fibrogenesis and Tumorigenesis

**DOI:** 10.3390/ijms23126852

**Published:** 2022-06-20

**Authors:** Moon Kee Meang, Saesbyeol Kim, Ik-Hwan Kim, Han-Soo Kim, Byung-Soo Youn

**Affiliations:** 1Osteoneurogen. Inc., Seoul 08501, Korea; moonkee@osteoneurogen.com (M.K.M.); byeol@osteoneurogen.com (S.K.); 2Department of Biotechnology, Korea University, Seoul 02841, Korea; ihkkim@korea.ac.kr; 3Department of Biomedical Sciences, College of Medical Convergence, Catholic Kwandong University, Gangneung-si 25601, Gangwon-do, Korea; hankim63@gmail.com; 4Basic Research Division, Biomedical Institute of Mycological Resource, College of Medicine, Catholic Kwandong University, Gangneung-si 25601, Gangwon-do, Korea

**Keywords:** cellular senescence, senolytics, cancer cell senolytics (CCS), small molecule

## Abstract

Uncontrolled proliferative diseases, such as fibrosis or cancer, can be fatal. We previously found that a compound containing the chromone scaffold (CS), ONG41008, had potent antifibrogenic effects associated with EMT or cell-cycle control resembling tumorigenesis. We investigated the effects of ONG41008 on tumor cells and compared these effects with those in pathogenic myofibroblasts. Stimulation of A549 (lung carcinoma epithelial cells) or PANC1 (pancreatic ductal carcinoma cells) with ONG41008 resulted in robust cellular senescence, indicating that dysregulated cell proliferation is common to fibrotic cells and tumor cells. The senescence was followed by multinucleation, a manifestation of mitotic slippage. There was significant upregulation of expression and rapid nuclear translocation of p-TP53 and p16 in the treated cancer cells, which thereafter died after 72 h confirmed by 6 day live imaging. ONG41008 exhibited a comparable senogenic potential to that of dasatinib. Interestingly, ONG41008 was only able to activate caspase-3, 7 in comparison with quercetin and fisetin, also containing CS in PANC1. ONG41008 did not seem to be essentially toxic to normal human lung fibroblasts or primary prostate epithelial cells, suggesting ONG41008 can distinguish the intracellular microenvironment between normal cells and aged or diseased cells. This effect might occur as a result of the increased NAD/NADH ratio, because ONG41008 restored this important metabolic ratio in cancer cells. Taken together, this is the first study to demonstrate that a small molecule can arrest uncontrolled proliferation during fibrogenesis or tumorigenesis via both senogenic and senolytic potential. ONG41008 could be a potential drug for a broad range of fibrotic or tumorigenic diseases.

## 1. Introduction

Uncontrolled progression of the cell cycle is linked to the development of fibrosis and cancer [1]. The entry of cells into the cell cycle and exit from the cell cycle needs to be tightly regulated [2], otherwise tumorigenesis can occur [3]. Although targeting cell-cycle progression using small molecules is a potential anticancer strategy, it may be associated with undesirable risks such as elimination of bystander noncancerous cells, systemic inhibition of stem-cell differentiation, or aggravation of homeostatic immunity [4]. Compounds that contain chemical scaffolds found in natural products, such as the chromone scaffold (CS) found largely in flavones or isoflavones, might be useful antiproliferative drugs; these natural products are largely safe and suppress uncontrolled proliferation as shown in studies using pathogenic myofibroblasts or aggressive tumor cells [5]. Metastasis plays a key role in the additional development of cancer and, thus, targeting metastasis is an active area of research [6]. It is increasingly clear that metabolic regulation in cancer cells is closely coupled to the progression of the cell cycle. Moreover, it is well established that tumor cell growth is exclusively dependent on aerobic glycolysis (also known as lactate fermentation) [7]. Similarly, pathogenic myofibroblasts utilize aerobic glycolysis and tend to generate a hypoxic environment; thus, aerobic glycolysis also plays a central role in the initiation and perpetuation of fibrosis [8].

Flavones are members of the polyphenol family of compounds, which contains over 10,000 compounds exclusively found in the plant kingdom [9]. In general, these phytochemicals protect plants from radiation damage [10]. Due to the fact of their anti-oxidant or anti-inflammatory potential, flavones have long been used to treat inflammatory diseases such as arthritis and asthma [11]. Chromone, 1,4-benzopyrone-4-one, is a central chemical scaffold that is found in flavones and isoflavones [12]. We recently reported that eupatilin, a CS-containing compound isolated from an *Artemisia* species, and its synthetic analog ONG41008 inhibit fibrogenesis in vitro and fibrosis in vivo. These effects occurred as a result of actin depolymerization followed by disassembly of the latent TGFβ complex (LTC), resulting in inhibition of epithelial–mesenchymal transition (EMT) [13]. Here, we show that ONG41008 potently induces cellular senescence in pathogenic myofibroblasts and oncogene-induced senescence (OIS)/cancer cell senolysis in several cancer cell lines being representative of aggressive forms of human cancer, resulting in cellular escape from fibrogenesis and tumorigenesis. These antiproliferative actions of ONG41008 suggest it could be a new therapeutic modality for treating fibrotic diseases as well as cancers.

## 2. Results

### 2.1. ONG41008 Induced Cellular Senescence in DHLFs

Diseased human lung fibroblasts (DHLFs) from idiopathy pulmonary fibrosis (IPF) patients are αSMA+ pathogenic myofibroblasts that express a range of muscular collagens. We sought to identify if the mechanism of action of ONG41008 was different from that of two first-in-class IPF drugs: nintedanib and pirfenidone. Nintedanib, which has high cellular toxicity, was originally developed as an anticancer drug and has been repurposed as an anti-IPF drug. Pirfenidone is hepatotoxic at high concentrations and indirectly modulates the function of immune cells, resulting in reduced activity of TGFβ [14,15]. We recently reported that ONG41008 inhibited TGFβ biogenesis, thereby blocking TGFR signaling and reprogramming the EMT [13]. In the current study, as shown in Figure 1A, nintedanib induced robust cell death in DHLFsat 24 h, and after a longer exposure of 48 h, nearly all DHLFswere dead. The IC_50_ value of nintedanib was 10.17 μM at 72 h. Pirfenidone had no effect on the survival of DHLFs. Interestingly, ONG41008 partially affected cell survival; the cell survival rate was more than 60%, even when concentrations of ONG41008 were increased, meaning we were unable to obtain an IC_50_ value. This result strongly indicates that ONG41008 treatment caused systemic growth arrest of DHLFs. As a control experiment, normal human lung fibroblasts (NHLFs) were treated with nintedanib, pirfenidone, and ONG41008. Surprisingly, although nintedanib highly inhibited cell survival, as in the case of DHLFs (the IC_50_ value was approximately 30 μM; Figure 1B), ONG41008 did not affect the survival of NHLFs. The same held true for pirfenidone. This finding meant we were unable to calculate an IC_50_ value for ONG41008 in NHLFs either.

To rule out the involvement of apoptosis in ONG41008-mediated growth arrest in DHLFs, three apoptotic features were measured using (1) an activated caspase-3 assay, (2) measurement of the mitochondrial membrane potential (MTMP), and (3) an LDH cytotoxicity assay. FCCP (trifluoromethoxy carbonylcyanide phenylhydrazone) was used as a control in the MTMP experiments. As shown in Figure 2A–C, only nintedanib caused significant apoptosis of DHLFs in the three assays, allowing estimation of respective IC_50_ or EC_50_ values. ONG41008 and pirfenidone were unable to elicit pro-apoptotic pathways. This result suggests that the systemic growth arrest induced by ONG41008 in DHLFs was not a consequence of apoptosis and was, instead, likely related to cellular senescence.

### 2.2. ONG41008 Induced Cellular Senescence Similarly to Dasatinib in DHLFs

To further explore cellular senescence in DHLFs, the cells were stimulated with varied concentrations of ONG41008 at different incubation time points as shown in Figure 3A. A hallmark of cellular senescence is cell flatness [16]. We used time-lapse microscopy to monitor the effects of increasing concentrations of ONG41008 in DHLFs. Replicative senescence occurred at 5 μM, and the senescence induction degree of ONG41008 became more evident at 10 μM ONG41008, with the maximal senescence occurring at 20 μM ONG41008.

Senogenic or senolytic compounds were first identified in 2017 [17,18,19]. Dasatinib, a Bcr-Abl and Src kinase inhibitor, was the first clinically used drug that induced cellular senescence. Quercetin and fisetin were pioneering senolytic drugs [20,21]. Interestingly, like quercetin and fisetin, ONG41008 contains a CS [22]. We previously reported that three compounds containing a CS had potent antifibrogenic activity [13], of which ONG41008 had the highest antifibrogenic capability. For comparing senogenic potential between ONG41008 and dasatinib, DHLFs were stimulated with 20 μM ONG41008 or 20 μM dasatinib for 72 h, and p16 expression levels were measured by a fluorescence analyzer (Cellest 5, Invitrogen, Waltham, MA, USA). While dasatinib gave rise to higher p16 fluorescence at 24 or 48 h, ONG41008 showed a maximal fluorescence at 72 h. Decreasing fluorescence by dasatinib might be due to the fact of cellular toxicity, whereas ONG41008 was not harmful to cells (Figure 3B). H2AX is a senescence-specific histone [23]. To further substantiate ONG41008-mediated RS, DHLFs were stimulated with 20 μM ONG41008 for 24, 48, or 72 h and stained with anti-H2AX. As shown in Figure 3C, the H2AX were nuclear localized and heavily stained, suggesting that ONG41008 is a potent inducer of replicative senescence. Video analysis also showed that 10 μM ONG41008 strongly induced replicative senescence (Appendix A Video S1). Nintedanib eliminated DHLFs.

### 2.3. TP53, p21, and p16 Activities Were Required for ONG41008-Induced Cellular Senescence in DHLFs

Several tumor suppressor proteins are responsible for halting the progression of the cell cycle [24]. It has been well reported that TP53 is phosphorylated at multiple sites and coupled mainly to p21 or p16, leading to cell-cycle inhibition. As shown in Figure 4A, 1 μM ONG41008 treatment did not alter the expression of TP53 but caused rapid translocation of this to the nucleus. Treatment of DHLFs with 1 to 10 μM ONG41008 upregulated p21 expression and caused translocation of the protein to the nucleus (Figure 4B,C). ONG41008 did not induce phosphorylation of ATM or p38, suggesting that ONG41008 is not a genotoxic chemical (data not shown).

### 2.4. ONG41008 Generated Several Potential Interactomes in DHLFs during Transdifferentiation into RS

Next, we conducted an RNA-seq analysis, giving rise to the nuclear interactome shown in Figure 4D. EGR1 (early growth response 1) seemed to be a priming protein, which interacted with proteins including CDC45 (cell division cycle 45), TACC3 (transforming acidic coiled-coil-containing protein 3), and CDK1 (cyclin-dependent kinase 1). These proteins play pivotal roles in the onset of nuclear reprogramming for trans-cell differentiation, cell-cycle control, and cellular senescence [25,26,27,28]. We also discovered that unique transcriptomes are responsible for driving RS; RNA-seq analysis was conducted to compare differences in gene expression between untreated DHLFs and 10 μM ONG41008-treated DHLFs. Upregulated genes were sorted according to their *p*-values. As shown in Appendix A Appendix A, three interactomes seemed to be working cooperatively when cells undergo replicative senescence. The first was a metabolic interactome typified by pyruvate dehydrogenase kinase (PDK)1, the second was actin biogenesis, and the last was related to histone modification, DNA replication, and the generation of a muscle–neuron signature. The reason why a muscle–neuron signature is generated for replicative senescence remains to be explored. Interestingly, the vast majority of the top-ranked genes affected by ONG41008 were soluble factors or receptors (Appendix A Appendix A). We speculate that to drive replicative senescence, expression of these cytokines or their interaction with cognate receptors is required to prime cellular senescence prior to the formation of euchromatin formation.

### 2.5. ONG41008 Harbors Cancer-Cell Senogenic Potential

Figure 5 summarizes the antifibrogenic, senogenic, and senolytic properties associated with the chromone scaffold-containing chemical structures. It has been shown previously that the presence of a methoxy group at the C6 position in the CS is essential to mediate antifibrogenic effects and, as such, apigenin, quercetin, and fisetin were not antifibrogenic. An immunocytochemistry (ICC) experiment showed that robust cellular senescence occurred when A549 cells were stimulated with ONG41008, and MNCs were clearly formed (Figure 6A). MNC is an early signature for mitotic slippage followed by senolysis [29]. It has been well documented that cellular senescence pertinent to cancer cells refers to oncogene-induced senescence (OIS) [30]. ONG41008 was antifibrogenic, senogenic, and senolytic. However, chemical analogs of ONG41008, namely, hispidulin, jaceosidin, and apigenin, did not induce cellular senescence. A549 cells were stimulated with ONG41008, quercetin, fisetin, or dasatinib (as a senogenic control). ONG41008, quercetin, and fisetin induced senolysis that was characterized by MNC formation (Figure 6B, highlighted by white circles). Dasatinib was toxic to A549 cells. In particular, A549 became rapidly senescent when treated with 10 μM ONG41008. Although fisetin appeared to induce senescence, but when treated with 20 μM quercetin or fisetin, the flat cell morphology disappeared and even appeared shrunken under higher magnification. No MNCs were observed in DHLFs stimulated with ONG41008, and we were unable to confirm senolysis of DHLFs stimulated with ONG41008, which remains to further be elucidated.

Taken together, ONG41008 is a potent inducer of cellular senescence in cancer cells and contains CS but is distinguished from quercetin and fisetin, which are generally known to be senolytics.

### 2.6. ONG41008 Is a Cancer Cell Senolytic

To quantitate the senolytic activity of ONG41008, quercetin, and fisetin containing a CS, CCK-8 assays used for early-phase detection of apoptosis were conducted by using PANC1 cells, and the IC_50_ values of these compounds were compared as shown in Figure 7A. Cisplatin was used as an apoptosis control. Quercetin exhibited the highest senolytic activity, and ONG41008 and fisetin also showed significant senolytic activity. The finding that quercetin and fisetin showed senolytic activity indicates that senescence may have occurred in PANC1 cells. Caspase-3 and -7 assays may be a good tool for determining late-stage senolysis. Interestingly, ONG41008 only exhibited a decent senolytic activity in terms of activation of caspase-3 and -7 in conjunction with dasatinib and cisplatin (Figure 7B). Further studies are needed to determine the effects of these compounds on cellular senescence and senolysis in more cancer cell lines and human primary cells. We established 6 day live imaging to ensure that the cancer cell senolytics was conceptually sound and ONG41008 completely killed PANC1 through senolysis (Appendix A). PANC1 cells turned into a growth retardation stage after 24 h of ONG41008 treatment and underwent massive cell death after 48 to 72 h. Cisplatin was used for control. In addition to A549, ONG41008-mediated cancer cell senolytics was explored in malignant human cancer cell lines: PANC1 cells, the human triple-negative (TNBC) breast cancer cell line MCF7, and PC3 cells (an MDR+/multidrug-resistant human prostate cancer cell line). These cell lines were stimulated with ONG41008 for 48 h and ICC was conducted. All three cancer cell lines showed robust cellular senescence as shown in Appendix A Appendix A. MNCs are denoted by white circles. In multidrug resistance, it has been well appreciated that cancer cells escape from drug targets [31]. Especially, human metastatic prostate cancer cells acquire this drug escape ability [32]. Since cancer cell senolytics would be based on recognition of cancer cell senescence rather than elicitation of genotoxic response, it could be possible that this new modality may be able to nullify or attenuate multidrug resistance. PC3 cells were stimulated with cisplatin or ONG41008 and 6 day live imaging was conducted. As shown in Appendix A, while PC3 stimulated with 5 or 10 μM cisplatin were significantly resistant to cisplatin, stimulation of PC3 cells with 20 μM ONG41008 effectively killed cancer cells, suggesting that cancer cell senolytics may attenuate or nullify multidrug resistance.

Taken together, these results show that ONG41008 is both senogenic and senolytic as well as antifibrogenic, suggesting that the therapeutic potential of ONG41008 could be extended to many age-associated diseases.

### 2.7. Biochemical Analysis of ONG41008-Mediated Cancer Cell Senolytics in A549

Pathogenic myofibroblasts and tumor cells share several features. First, aerobic glycolysis is the major pathway for the catabolism of glucose. Second, uncontrolled cell division occurs. Third, their survival is dependent on immune escape. The observation that ONG41008 rapidly caused DHLFs to become replicative senescence prompted us to see if cancer cells would behave similarly to DHLFs. Cellular senescence was established in A549 cells by treatment with 10 μM ONG41008 for 24, 48, or 72 h. Similar to our studies of replicative senescence in DHLFs, we explored changes in the expression and location of TP53, p21, and p16, which can induce RS in A549 cells. As shown in Appendix A Appendix A, TP53 protein expression remained unchanged during ONG41008 treatment, but 1 μM ONG4008 caused nuclear translocation of TP53. Expression of p21 was upregulated and similarly localized in the nucleus (Appendix A Appendix A). Expression of p16 was concentration-dependently upregulated by ONG41008 and was localized in the nucleus and distributed at the perinuclear zones (Appendix A Appendix A). We discovered that the cell morphology was more homogenous 72 h after ONG41008 treatment than 24 h after treatment; this result may reflect the elimination of multinucleated A549 cells. Moreover, maximal translocation of p16 to the nucleus was complete at 72 h. We anticipate that p16-saturated A549 cells would undergo senolysis (Appendix A Appendix A). Since the ability of TP53 to regulate cell-cycle arrest is dependent on its phosphorylation of [33], we conducted western blot analysis using a phospho-specific TP53 antibody. As shown in Appendix A Appendix A, ONG41008 concentration-dependently increased the level of TP53 phosphorylation; the level of total TP53 remained unchanged. ONG41008 also induced expression of p21 and p16 in a concentration-dependent manner; this finding corroborated the results of ICC experiments. Next, we treated A549 cells with TGFβ to increase proliferation in the presence or absence of ONG41008. A significant proportion of A549 cells died off after 6 days of treatment with ONG41008 and all cells died after 15 days of treatment, whereas control A549 cells remained alive (Figure 6B).

ONG41008 and ONG21001 have senogenic and senolytic effects. No antifibrogenic capacities and senogenic activity were associated with the structure quercetin and fisetin. 

To further explore the senolysis induced by ONG41008, PANC1 cells were employed and treated with ONG41008. To verify if cellular senescence or senolysis occurred, Western blot analyses were conducted, and the results are shown in Figure 8A. Induction of p-TP53, p-Rb, and caspase-3, Mcl1, and cleavage of PARP were evident, suggesting that ONG41008 induced senolysis in PANC1 cells. Furthermore, ICC experiments showed that induction of p-TP53, p21, p16, Mcl1, and p-Rb were notable and rapid translocation to nucleus occurred (Figure 8B). Interestingly, induction of Mcl1, an anti-apoptotic Bcl2 family protein, caught our attention because induction of Mcl1 seemed to be a signature induced protein in OIS [29].

Taken together, these results show that ONG41008 induced cellular senescence in cancer cells followed by senolysis via heavy G2/M arrest, mitotic collapse, multinucleation, and induction of Mcl1. We would like to propose that this kind of anticancer modality be defined as “Cancer Cell Synolytics (CCS)”.

### 2.8. Recognition of the Intracellular Microenvironment by ONG41008 May Determine Strength of Cancer Cell Synolytics (CCS)

In order for us to confirm CCS as an intrinsic cancer cell program, a total of eleven established cancer cell lines or primary NSCLCs and SCLCs were tested. Although a wide degree of CCS responsiveness was noted, without exception, all cancer cells contacted with ONG41008 exhibited robust cell growth arrest, namely, cancer cell senescence followed by G2/M arrest, with multinucleation being an initial feature of mitotic collapse. In addition, massive cell death was initiated. As described above, ONG41008 induced cellular senescence in DHLFs. Especially, ONG41008 did not induce senolysis in DHLFs. However, nintedanib induced robust apoptosis in DHLFs, prompting us to hypothesize that ONG41008 might differently sense pathogenic or aged-cell microenvironments but not affect homeostatic intracellular microenvironments. This hypothesis is very challenging to test. First, we attempted to confirm if NHLFs were unresponsive to ONG41008 but underwent rapid cellular toxicity in response to cisplatin or nintedanib. As shown in Figure 9A,B, ONG41008 exerted no apoptotic effects on NHLFs, and it was not possible to obtain an IC_50_ value, whereas cisplatin and nintedanib killed NHLFs with IC_50_ values of 18.14 and 4.59 μM, respectively. We tested this hypothesis in human primary prostate epithelial cells (HPrECs). In these cells, the IC_50_ value of ONG41008 was 46.77 μM and the IC_50_ value of cisplatin was 0.14 μM, indicating that ONG41008 did not much impact HPrECs in terms of apoptosis (Figure 9C,D). In addition, we employed 6 day live imaging to determine if varying concentrations of ONG41008 caused harmful effects on NHLFs (normal human lung fibroblasts) and HPrCEs (human prostate primary epithelial cells). As shown in Appendix A Video S4 no harmful effects were noted, confirming the reliability of the senolysis assays as described above.

Taken together, these results indicate that unlike cisplatin, ONG41008 exerted apoptotic effects on both A549 and PANC1 cells by recognizing a certain critical level of senescence stemming from cancer cell intrinsic senescence plus the senogenic potential associated with ONG41008. At this point, ONG41008 acted as CCS, leading to senolysis, which eventually resulted in permanent cell death. However, normal cells or young cells were not affected by ONG41008.

### 2.9. ONG41008 Induced Cell-Cycle Arrest at G2/M and Restored the NAD/NADH Ratio 

The involvement of TP53, p16, and p21 in the action of ONG41008 strongly indicates that ONG41008 plays an important role in controlling the cell cycle. We conducted cell-cycle analysis using PI staining 24 and 48 h after treatment of A549 cells with ONG41008 or fisetin as a control. ONG41008 as well as fisetin induced cell-cycle arrest at the G2/M stage (Figure 10A). Abrogation of CDK2 and CDK6 activity by ONG41008 could be further evidence for G2/M arrest (Figure 10B). Metabolic regulation is a key driver of cell fate, determining if the cell cycle is controlled or whether cells undergo apoptosis [34]. It is well established that the NAD/NADH ratio plays a central role in regulating energy metabolism, including the control of glycolysis and the TCA cycle, and substantially affects mitochondrial functions such as those involved in disease pathogenesis or aging [35]. We hypothesized that ONG41008 could influence the NAD/NADH ratio, based on the hitherto observation that ONG41008 could “sense” the intracellular microenvironment and so distinguish normal cells from those undergoing uncontrolled proliferation, such as tumor cells or pathogenic myofibroblasts. When A549 cells were stimulated with ONG41008, the NAD/NADH ratio increased to between 60 and 80 (Figure 10C). However, SAHA did not affect the NAD/NADH ratio. In the absence of ONG41008, the NAD/NADH ratio in A549 cells had a negative value, suggesting that A549 cells may exclusively utilize anaerobic glycolysis when they are exposed to ONG41008. Due to the variability in measuring concentrations of NADH by a commercial NAD/NADH ratio-measuring assay, we performed a total of six experiments. However, the tendency of the increase in NAD upon ONG41008 treatment seemed evident. PANC1 cells similarly behaved (data not shown).

## 3. Discussion

Fibrosis and cancer are often intractable and fatal diseases with a pathogenesis that is directly related to cell-cycle regulation. Under normal conditions, the cell cycle is meticulously controlled such that proliferation, differentiation, and apoptosis are coordinated. Dysfunction of this tight regulation, which leads to uncontrolled proliferation, may increase the likelihood of serious diseases. Currently used antifibrotic and anticancer drugs largely target uncontrolled proliferation [36]. Yet such drugs are associated with several side effects. Immune checkpoint inhibitors, which enable the host immune system to destroy cancer cells, are becoming a very successful anticancer strategy and are associated with fewer side effects than drugs that target proliferation [37]. Here, we show that replicative senescence and OIS (senolysis) induced by ONG41008 caused the proliferative arrest of DHLFs and A549 cells, thereby preventing uncontrolled proliferation. Although we have not tested the effects of ONG41008 in many normal (that is noncancer and nonfibrotic) cell lines, the compound was clearly able to distinguish pathogenic myofibroblasts and several cancer cells from normal lung fibroblasts and primary prostate epithelial cells.

Pathogenic myofibroblasts and cancer cells both gain ATP from aerobic glycolysis, resulting in a dysregulated redox potential that manifests as alterations in the NAD/NADH ratio, ATP levels, or intracellular pH. The CS is found in flavones or isoflavones [38], and captures UV to protect plants from UV damage. Although over 10,000 CS derivatives have been identified, only a limited number have antifibrotic effects. This finding suggests that in addition to the CS, a variety of side chains linked to the CS are needed for the compound to have antifibrotic effects. Another study reported that the CS-like drug contained with the structure of SB203580 is important in mediating the anticancer activity of this compound [39]. Since ONG41008 can distinguish pathogenic myofibroblasts (DHLF) and cancer cells (A549, PANC1, MCF7, and PC3 cells) from normal cells, we propose that the CS might recognize features of the intracellular microenvironment, such as NAD, NADH, pH, or ATP, that act as energy sensors. As ONG41008 normalized the NAD/NADH ratio to a value of 60–100, we speculate that the normalized NAD/NADH ratio may influence ROS production in ONG41008-treated A549 cells, thereby partly contributing to senolysis. Because ONG41008 increases intracellular NAD concentrations, it could also be possible that the NAD rate-limiting enzyme, NAMPT (nicotinamide phosphoribosyltransferase), was the molecular target of ONG41008.

Overall, our findings indicate that ONG41008 is a potent inducer of cellular senescence and senolysis, which arrests the pathogenic proliferation of myofibroblasts or cancer cells, thus preventing uncontrolled proliferation.

## 4. Materials and Methods

### 4.1. Cell Culture and Reagents

DHLFs were purchased from Lonza (Basel, Switzerland) and cultured in fibroblast growth medium (FBM, Lonza, Walkersville, MD, USA). Recombinant human TGFβ and PDGF were obtained from Peprotech (Rocky Hill, CT, USA) and used at a final concentration of 5 ng/ml. Chemically synthesized ONG41008 was obtained from Syngene International Ltd. (Bangalore, India), dissolved at a stock concentration of 50 mM in DMSO, and stored in aliquots at −20 °C. DMSO was used as a control. The RAW264.7 cell line was purchased from the Korean Cell Line Bank (Seoul, Korea) and cultured in RPMI supplemented with 10% FBS and 1% P/S (Welgene, Seoul, Korea). LPS was purchased from Sigma and used at a final concentration of 100 ng/mL.

### 4.2. Immunocytochemistry

Cells were fixed using 4% paraformaldehyde, permeabilized with 0.4% TritonX100, blocked with 1% BSA, and incubated with rhodamine phalloidin (Thermo Fisher, Waltham, MA, USA), anti-GATA6 (Abcam, Cambridge, UK), anti-p53 (Cell Signaling Technology, Beverly, MA, USA), p21(Abcam, Cambridge, UK), p16-INK4A (Proteintech, Rosemont, IL, USA), and ZEB1 (Cell Signaling Technology, Beverly, MA, USA) for 4 h at room temperature. After washing, cells were incubated with an Alexa Fluor 488 (Abcam, Cambridge, UK)-conjugated secondary antibody. Images were analyzed using EVOS M7000 (Invitrogen, Waltham, MA, USA).

### 4.3. Western Blotting

The A549 cells were seeded at 1 × 10^6^ cells/well in 100 mm cell culture dishes and incubated overnight, followed by treatment with various concentrations of ONG41008. After 24 h, the cell lysates were clarified by centrifugation at 14,000× *g* for 10 min and the supernatant was collected. The protein concentrations were quantified by the Bradford assay (Thermo Fisher, Waltham, MA, USA). Thereafter, 25 µg of cellular protein was loaded on a 10% SDS-PAGE gel and transferred to nitrocellulose membranes. After blocking with 5% BSA, the membranes were incubated with anti-p53 (Cell Signaling Technology, Beverly, MA, USA), phosphor-p53 (Cell Signaling Technology, Beverly, MA, USA), p21 (Abcam, Cambridge, UK), p16-INK4A (Proteintech, Rosemont, IL, USA), and GAPDH (Abcam, Cambridge, UK) overnight at 4 °C. After washing thoroughly, membranes were incubated with an HRP-conjugated secondary antibody. Protein bands were visualized using ECL reagent (Abfrontier, Seoul, Korea) and Uvitec HD9 (Uvitec, Cambridge, UK).

### 4.4. Live Imaging

DHLFs were seeded in 12-well cell culture plates and, after 24 h, were treated with TGF beta (5 ng/ml), nintedanib (10 µM), pirfenidone (10 µM), and ONG41008 (10 µM). Cells were incubated in an EVOS M7000 CO_2_ incubation chamber (Invitrogen, Waltham, MA, USA); cell morphology images were captured every 30 min. The collected images were assembled and are available as video (Appendix A Information).

### 4.5. CCK-8 Assay

Cells were seeded onto 96-well plates at a density of 3 × 10^4^ cells/well for 24 h. Then, cells were treated with 0.01, 0.1, 1, 10, 20, and 30 µM ONG41008, pirfenidone, or nintedanib. After treatment, the cell survival rate was measured using a CCK-8 assay (Dojindo, CK04, Rockville, MD, USA) according to the manufacturer’s protocol.

### 4.6. Caspase-3 Assay

The caspase-3 activity was measured using a caspase-3 assay kit (Abcam, ab37401) following the manufacturer’s protocol. Cells treated with various concentrations of ONG41008, pirfenidone, or nintedanib were harvested and lysed on ice. The protein concentration was then measured by a BCA assay (ThermoFisher, 23227) and adjusted to 50 µg protein per 50 µL cell lysis buffer.

### 4.7. Mitochondrial Membrane Potential Assay

Cells were seeded in a 96-well plate for 24 h, then exposed to different concentrations of ONG41008, pirfenidone, or nintedanib. The cells were then co-incubated with TMRE (Abcam, ab113852) for 30 min at 37 ℃ in the dark. Cells treated with 20 µM FCCP were used as a positive control. The mitochondrial membrane potential was measured following the manufacturer’s protocol.

### 4.8. LDH Assay

LDH release was detected using an LDH assay kit (Abcam, ab56393). The cell culture plates were centrifuged at 480× *g* for 10 min, and supernatants (10 µL/well) were extracted into a new 96-well plate. Then, 100 µL of LDG reaction mix was added to each well and incubated for 30 min at room temperature. The absorbance values were measured at 450 nm on a microplate reader.

### 4.9. NAD/NADH Assay

Total NAD was extracted and quantified from A549 cell lysates using an NAD+/NADH colorimetric assay kit (Abcam, ab65348) following the manufacturer’s instructions. Cells (1 × 10^6^) were lysed in 400 µL NAD/NADH extraction buffer, filtered through a 10 kD spin column (ab93349) and measured neat or at a 1/5 dilution. Briefly, the amount of total NAD was calculated from a standard curve (pmol) divided by the sample volume added to the reaction well (µL) and multiplied by the dilution factor.

### 4.10. RNA-seq, Differential Gene Expression, and Interactome Analyses

Processed reads were mapped to the *Mus musculus* reference genome (Ensembl 77) using Tophat and Cufflink with default parameters. Differential analysis was performed using Cuffdiff using default parameters. Then, FPKM values from Cuffdiff were normalized and quantitated using the R Package tag count comparison (TCC) to determine statistical significance (e.g., *p*-values) and differential expression (e.g., fold-changes). Gene expression values were plotted in various ways (i.e., scatter, MA, and volcano plots) using fold-change values and an R-script developed in-house. The protein interaction transfer procedure was performed using the STRING database with the differentially expressed genes. A 60 Gb sequence was generated, and 10,020 transcripts were read and compared. The highest confidence interaction score (0.9) was applied from the *Mus musculus* reference genome, and information regarding interactions was obtained based on text mining, experiments, and databases (http://www.string-db.org/ accessed on 6 December 2021). Due to the proprietary company information, we have not provided a detailed interpretation of the RNA-Seq or interactome data, but we have provided sufficient analysis to support our assertions in Section 3 and Section 4.

### 4.11. Reverse Transcriptase PCR and Real-Time PCR

Cells cultured in either 12- or 24-well plates were washed twice with cold PBS and harvested using a TaKaRa MiniBEST universal RNA extraction kit (Takara, Japan). RNA was purified using the same kit according to the manufacturer’s protocol. RNA was reverse-transcribed using a cDNA synthesis kit (PCRBio Systems, London, UK). Synthesized cDNA was amplified with StepOne Plus (Applied Biosystems, Life Technologies, Waltham, MA, USA) and 2× qPCRBio probe mix Hi-ROX (PCRBio, London, UK). Comparisons between mRNA levels were performed using the ∆∆Ct method, with GAPDH as the internal control.

## 5. Layman’s Summary: The Research in Context

### 5.1. Evidence Prior to This Study

The notion that aging is a disease and that diseases occur as a consequence of aging was first put forward by David Sinclair and colleagues [40]. There are biological systems that provide evidence for this notion; for example, somatic cells can revert to embryonic cells, producing younger somatic cells. This phenomenon underlies induced pluripotent stem cells. Another example is that some types of jellyfish can live forever. These examples suggest that a counter-aging program exists in animals. Human diseases are the manifestations of cell aging generated by the accumulation of somatic mutations. Aged and pathogenic cells are senescent, so a drug that specifically targeted senescent cells might initiate a cellular program that could ameliorate age-associated disease. Indeed, the kinase inhibitor dasatinib induces cellular senescence [19]. In 2017, two drugs that target senescent cells were identified: quercetin and fisetin. These drugs selectively kill senescent cells and are referred to as senolytic drugs or senotherapeutics [41]. Although it is established that senescent cells accumulate in cancer and idiopathic pulmonary fibrosis [18], the effect of senolytic drugs on these diseases is largely unknown.

### 5.2. Added Value of This Study

This study characterized a novel drug, termed ONG41008, which was found to have both senogenic and senolytic effects in cell-based assays. ONG41008 induced senescence in myofibroblasts and several cancer cell lines representative of aggressive human cancers, which was followed by cell death. Importantly, ONG41008 exhibited essentially no toxicity on normal human lung fibroblasts or primary prostate epithelial cells.

### 5.3. Implication of All the Evidence

Based on our results, we believe that ONG41008 is a potent inducer of cellular senescence (replicative senescence and oncogene-induced senescence) and causes arrest of uncontrolled, pathogenic proliferation of myofibroblasts or cancer cells.

## Figures and Tables

**Figure 1 ijms-23-06852-f001:**
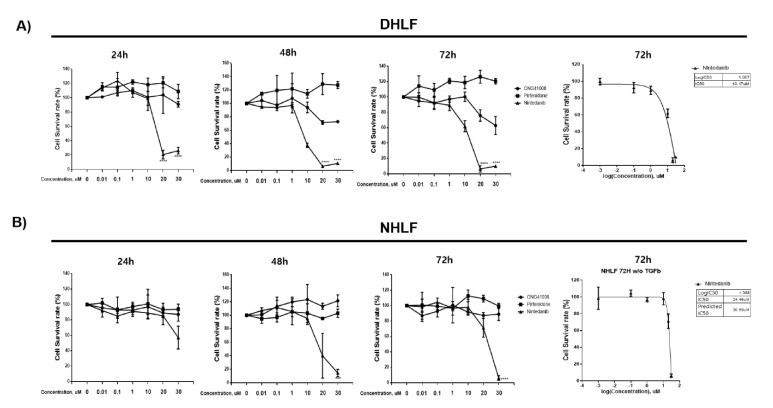
Comparison of the survival rate of DHLFs and NHLFs treated with ONG41008, nintedanib, or pirfenidone. (**A**) and (**B**) DHLFs and NHLFs were stimulated with various concentrations (up to 30 μM) of the indicated drugs for 24, 48, and 72 h. The cell survival rate was measured using a CCK-8 assay. The IC_50_ value of nintedanib was calculated with a sigmoidal, four-parameter logistic in GraphPad Prism, version 7.00. At least three independent measurements were conducted. Standard deviations are denoted by the bar scales.

**Figure 2 ijms-23-06852-f002:**
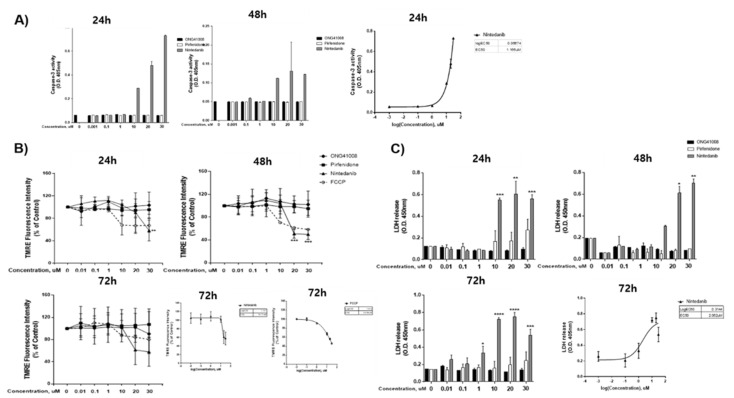
ONG41008 did not induce apoptosis in DHLFs. DHLFs were stimulated with ONG41008, pirfenidone, or nintedanib. (**A**) An activated caspase-3 assay was performed, and the EC50 value of nintedanib was calculated at 24 h. (**B**) The mitochondrial membrane potential was measured. The mitochondrial oxidative phosphorylation uncoupler FCCP was used as a control. The IC50 values of nintedanib and FCCP were calculated at 72 h. (**C**) Lactate dehydrogenase (LDH) levels were assayed. The EC50 value was calculated with a sigmoidal, four-parameter logistic in GraphPad Prism, version 7.00. At least three independent measurements were conducted. Standard deviations are represented by the bars. In Appendix A Video S1, 6 day live imaging clearly demonstrates the difference between ONG41008 and nintedanib. (* *p* < 0.01, ** *p* < 0.001, and *** *p* < 0.0001).

**Figure 3 ijms-23-06852-f003:**
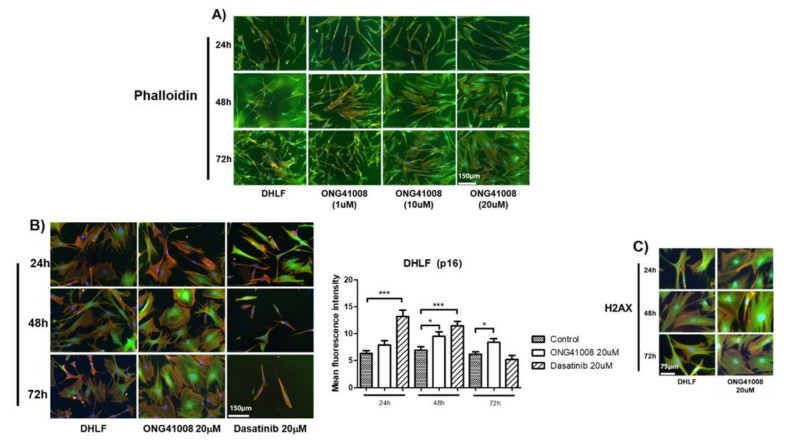
Induction of replicative senescence by ONG41008 and an interactome analysis. (**A**) DHLFs were treated with different concentrations of ONG41008 or a control (RPMI medium). Immunocytochemistry (ICC) was conducted with antihuman GATA6, phalloidin, and DAPI. Morphological changes were monitored under phase-contrast and fluorescence microscopy. Images that show typical replicative senescent cells are denoted by the phrase “replicative senescence”. (**B**) DHLFs were stimulated with 20 μM ONG41008 or 20 μM dasatinib for 24, 48, or 72 h, and replicative senescence was compared via ICC with the use of anti-p16. (**C**) DHLFs were stimulated for 24, 48, or 72 h and stained with anti-H2AX. ICC images were mounted.

**Figure 4 ijms-23-06852-f004:**
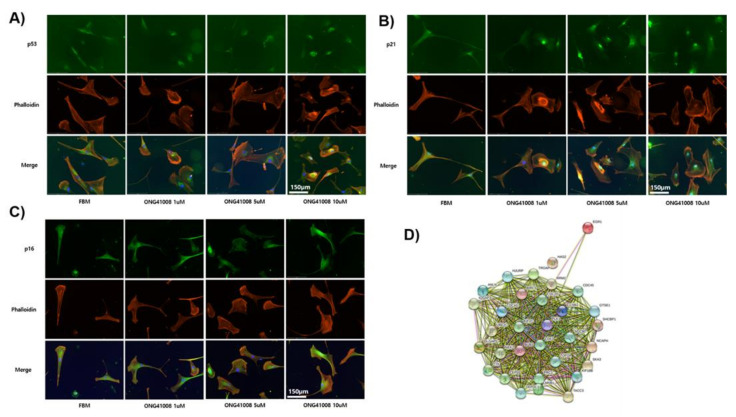
Translocation of TP53 to the nucleus, and induction and nuclear translocation of p21 and p16 in DHLFs upon ONG41008 treatment DHLFs were treated with 1 to 10 μM ONG41008 or a control (RPMI medium). ICC was conducted with anti-human TP53, anti-p21, anti-p16, or phalloidin in conjunction with DAPI. (**A**) Translocation of p53 to the nucleus; (**B**) induction and translocation of p21 to the nucleus; (**C**) induction and translocation of p16 to the nucleus; (**D**) before or after induction of replicative senescence RNA-seqs were conducted and a nuclear interactome was generated by the String program.

**Figure 5 ijms-23-06852-f005:**
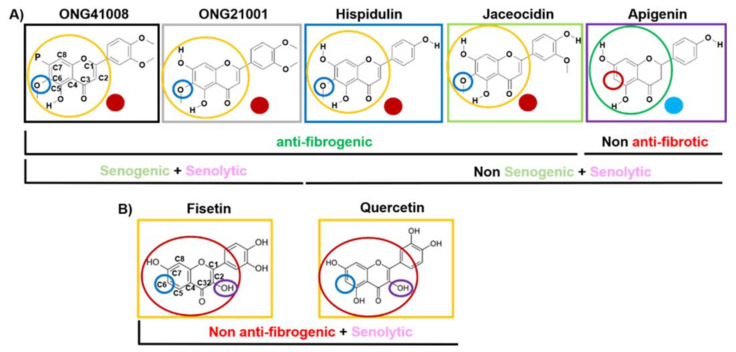
The chemical structure of compounds containing a chromone scaffold. (**A**) ONG41008, ONG21001, hispidulin, and jaceosidin, which are denoted by red circles, had antifibrogenic effects, and apigenin was not antifibrogenic, demonstrated by the blue circle. The presence of a methoxy group at C6 in the CS is necessary for antifibrogenic capacity. (**B**) The chemical structures for Fisetin and Quercetin) are represented.

**Figure 6 ijms-23-06852-f006:**
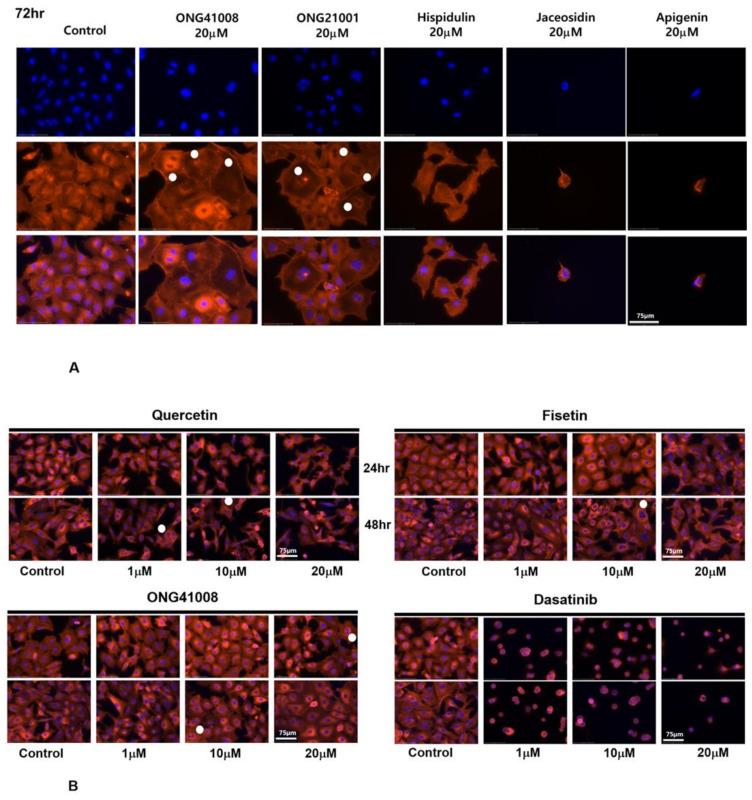
Cellular senescence and the senolytic capability: comparison between quercetin, fisetin, ONG41008, and dasatinib. (**A**) ONG41008 and ONG21001 induced cellular senescence leading to multinucleation. A549 cells were stimulated with 20 μM ONG41008, ONG21001, hispidulin, jaceosidin, or apigenin for 72 hr and subjected to ICC using DAPI and phalloidin. MNCs are denoted by white circles. (**B**) Comparison of the morphological changes upon stimulation of A549 with ONG41008, quercetin, fisetin, or dasatinib.

**Figure 7 ijms-23-06852-f007:**
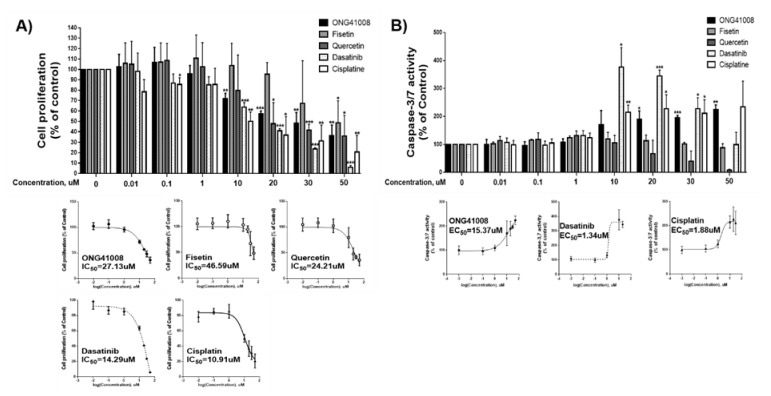
Comparison of senolytic capabilities. (**A**) PANC1 cells were treated with increasing concentrations (from 100 nM to 50 μM) of ONG41008, quercetin, fisetin, dasatinib, or cisplatin. CCK-8 assays were conducted, and IC50 values were calculated. (**B**) PANC1 cells were stimulated as described above and Caspase-3 and -7 assays were conducted. EC50 values were also calculated. Three independent experiments with triplicates were performed, and statistical acquisition was completed for ONG41008, fisetin, quercetin, and cisplatin (* *p* < 0.01, ** *p* < 0.001, and *** *p* < 0.0001).

**Figure 8 ijms-23-06852-f008:**
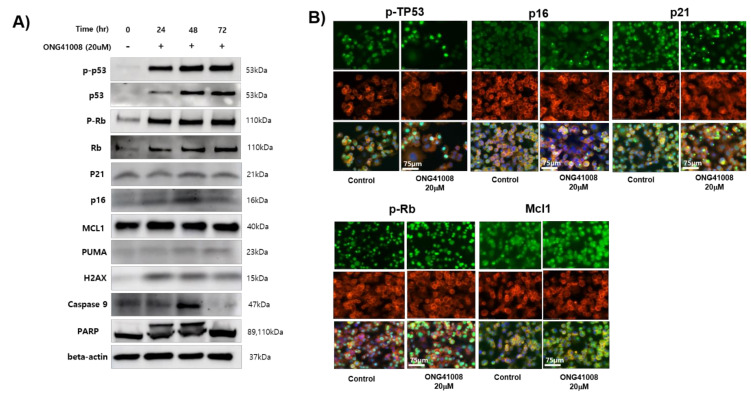
Biochemical and ICC analysis of cancer cell senolytics (CCS) associated with ONG41008. (**A**) PANC1 cells were stimulated with 10 μM ONG41008 for 6 h. ICC was conducted with antibodies against p-TP53, p21, p16, p-Rb, or Mcl1, giving rise to green cells and further stained with phalloidin (red) and DAPI (blue). (**B**) PANC1 cells were treated with 20 μM ONG41008 for 48 h. Western blot analysis was conducted to detect p53, phospho-p53, Rb, phospho-Rb, caspase-3, caspase-9, PARP, p21, p16, Mcl1, PUMA, and actin.

**Figure 9 ijms-23-06852-f009:**
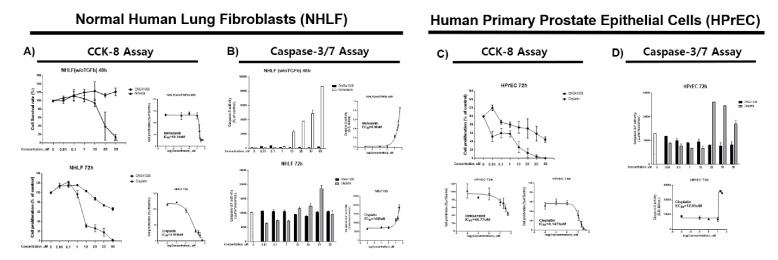
A comparison of the apoptotic effects of ONG41008, nintedanib, and cisplatin and the survival rate of NHLFs and HPrECs. (**A**) NHLFs were stimulated with ONG41008 or nintedanib for 48 h and stimulated with ONG41008 or cisplatin for 72 h. The cell survival rate was measured using a CCK-8 assay. The IC_50_ values of nintedanib and cisplatin were calculated. (**B**) NHLFs were stimulated with ONG41008 or nintedanib for 48 h and stimulated with ONG41008 or cisplatin for 72 h. An activated caspase-3 assay was performed, and the EC_50_ values of nintedanib and cisplatin were calculated. (**C**) Human prostate epithelial cells (HPrECs) were stimulated with ONG41008 or cisplatin for 72 h. The cell survival rate was measured using a CCK-8 assay. The IC_50_ values of ONG41008 and cisplatin were calculated. (**D**) HPrECs were stimulated with ONG41008 or cisplatin for 72 h. An activated caspase-3 assay was performed, and the EC_50_ value of cisplatin was calculated with a sigmoidal, four-parameter logistic in GraphPad Prism, version 7.00. The standard deviation was derived from three independent experiments. Appendix A Video S4 clearly showed that ONG41008 was not toxic to the normal primary human cells.

**Figure 10 ijms-23-06852-f010:**
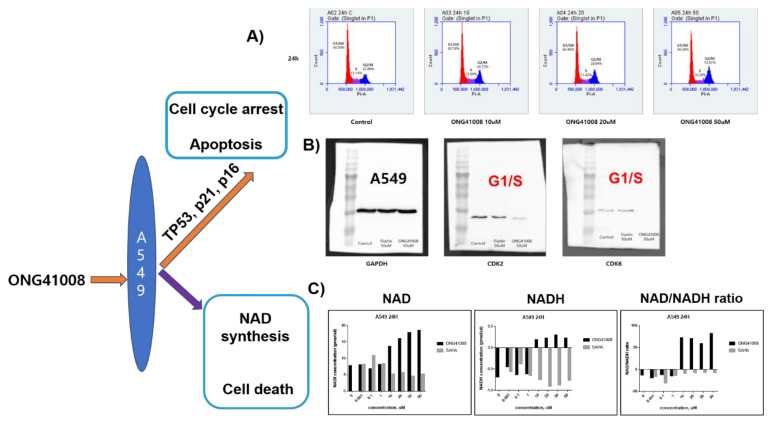
ONG41008-mediated cell-cycle arrest and induction of NAD/NADH and ROS expression (**A**) A549 cells were stained by PI. Cell cycle analysis was performed by FACS. (**B**) A549 cells were stimulated with 50 μM fisetin or ONG41008 for 24 h, and then cell lysates were collected. CDK2 and CDK6 expression was detected by Western blot analysis. (**C**) A549 cells were stimulated with medium alone, varying concentrations of SAHA or ONG41008 for 24 h. Cell lysates were prepared and the NAD/NADH ratio was measured. A representative NAD/NADH ratio was presented out of a total of six independent experiments.

## Data Availability

Not applicable.

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
