# Peer review of "A Small Molecule That Promotes Cellular Senescence Prevents Fibrogenesis and Tumorigenesis"

_ijms, 2022, doi:10.3390/ijms23126852_

Round 1

Reviewer 1 Report

This is a study of the senogenic and senolytic potential of compound ONG41008 on several cell lines representative of aggressive human cancers. The study is original, well designed, executed and presented. 

Please note the following two typos:

page 6 title: ONG41008 is a cancer cell senolytic NOT senolytics

page 11 last sentence: ....were not affected by ONG41008.

Author Response

  • Response to Reviewer 1

Thank you for your comments that will be improved.

<Comment>

page 6 title: ONG41008 is a cancer cell senolytic NOT senolytics

<Response>

We have corrected this sentence that is highlighted by a yellow shadow.

<Comment>

page 11 last sentence: ...were not affected by ONG41008.

<Response>

We have corrected conforming to the reviewer's comment that is highlighted by a yellow shadow.

We discovered several more typos and fixed them. These are highlighted by yellow shadows.

Reviewer 2 Report

Comment 1: The authors must review the references

Comment 2: The authors should put the scale bar

Comment 3: In the legend figures, the authors should include the stats

Author Response

  • Response to Reviewer 2

Thank you for your comments on which the present manuscript would be much improved.

<Comment 1>

The authors must review the references

<Response 1>

We have gone over the references and fixed the reference number. These changes were highlighted by blue shadow.

<Comment 2>

The authors should put the scale bar.

<Response 2>

The scale bars were inserted in all ICC figures.

<Comment 3>

In the legend figures, the authors should include the stats.

<Response>

We inserted descriptions of the standard deviations in Figure 2 and Figure 9. These data were vital because our 6-Day live imaging clearly exhibited cancer cell killing by ONG41008. Regarding Figure 10C, NAD/NADH ratio, the commercial assay kits gave us varying NADH concentrations upon ONG41008 treatment. However, the increase in NAD concentrations was consistent. We performed a total of six independent experiments. PANC1 cells similarly behaved. This statement was inserted into the corresponding text and highlighted by a blue shadow.  We think that we shall be looking for another NAD/NADH ratio-measuring assay kit.
